# Impact of tiny targets on *Glossina fuscipes quanzensis,* the primary vector of human African trypanosomiasis in the Democratic Republic of the Congo

**Inaki Tirados**[1], **Andrew Hope**[1], **Richard Selby**[1], **Fabrice Mpembele**[2], **Erick Mwamba Miaka**[2‡], **Marleen Boelaert**[3†], **Mike J. Lehane**[1], **Steve J. Torr**[1], **Michelle C. Stanton**[1,4]*

**1** Department of Vector Biology, Liverpool School of Tropical Medicine, Liverpool, United Kingdom, **2** Programme National de Lutte contre la Trypanosomiase Humaine Africaine, Kinshasa, Democratic Republic of the Congo, **3** Department of Public Health, Institute of Tropical Medicine, Antwerp, Belgium, **4** Centre for Health Informatics, Computing and Statistics, Lancaster Medical School, Lancaster University, United Kingdom

† Deceased.
‡ Unavailable to confirm authorship.
* michelle.stanton@lstmed.ac.uk

**Data Availability Statement:** All relevant data are within the manuscript and its Supporting Information files.

## Abstract

Over the past 20 years there has been a >95% reduction in the number of Gambian Human African trypanosomiasis (g-HAT) cases reported globally, largely as a result of large-scale active screening and treatment programmes. There are however still foci where the disease persists, particularly in parts of the Democratic Republic of the Congo (DRC). Additional control efforts such as tsetse control using Tiny Targets may therefore be required to achieve g-HAT elimination goals. The purpose of this study was to evaluate the impact of Tiny Targets within DRC. In 2015–2017, pre- and post-intervention tsetse abundance data were collected from 1,234 locations across three neighbouring Health Zones (Yasa Bonga, Mosango, Masi Manimba). Remotely sensed dry season data were combined with pre-intervention tsetse presence/absence data from 332 locations within a species distribution modelling framework to produce a habitat suitability map. The impact of Tiny Targets on the tsetse population was then evaluated by fitting a generalised linear mixed model to the relative fly abundance data collected from 889 post-intervention monitoring sites within Yasa Bonga, with habitat suitability, proximity to the intervention and intervention duration as covariates. Immediately following the introduction of the intervention, we observe a dramatic reduction in fly catches by > 85% (pre-intervention: 0.78 flies/trap/day, 95% CI 0.676–0.900; 3 month post-intervention: 0.11 flies/trap/day, 95% CI 0.070–0.153) which is sustained throughout the study period. Declines in catches were negatively associated with proximity to Tiny Targets, and while habitat suitability is positively associated with abundance its influence is reduced in the presence of the intervention. This study adds to the body of evidence demonstrating the impact of Tiny Targets on tsetse across a range of ecological settings, and further characterises the factors which

**Funding:** The Bill & Melinda Gates Foundation (www.gatesfoundation.org) grant OPP1155293 funded this study. The funders had no role in study design, data collection and analysis, decision to publish or preparation of the manuscript.

**Competing interests:** The authors have declared that no competing interests exist. Authors Marleen Boelaert and Erick Mwamba Miaka were unable to confirm their authorship contributions. On their behalf, the corresponding author has reported their contributions to the best of their knowledge.

modify its impact. The habitat suitability maps have the potential to guide the expansion of tsetse control activities in this area.

## Author summary

There have been large declines in the number of cases of sleeping sickness as a result of programmes that actively screen and treat the at-risk population. Additional control is needed in areas where the disease persists such as parts of the Democratic Republic of Congo (DRC). The disease is transmitted by tsetse flies, and reducing the tsetse population using Tiny Targets has been shown to control the disease in other countries. Extensive tsetse monitoring has been undertaken in one Health Zone in DRC where Tiny Targets have been deployed. We used these data to gain a better understanding of tsetse habitat, to produce habitat suitability maps, and to subsequently measure the impact of Tiny Targets on the tsetse population. We show that tsetse flies are largely found along rivers and surrounding densely vegetated habitat, with there being a positive relationship between habitat suitability and the number of flies caught. Once Tiny Targets were introduced, the number of flies caught in monitoring traps decreased by >85%, with habitat suitability at the trap location, and the proximity of the trap to the nearest Tiny Target influencing the size of the effect of the intervention. This study adds to the body of evidence demonstrating the impact of Tiny Targets on tsetse distribution in addition to providing information that can be used to guide the expansion of tsetse control activities in this area.

## Introduction

Human African trypanosomiasis (HAT), commonly called sleeping sickness, is a neglected tropical disease caused by sub-species of *Trypanosoma brucei* transmitted by tsetse flies (*Glossina*). In Central and West Africa, HAT is caused by *T. b. gambiense* (Gambian HAT, g-HAT) transmitted by riverine species of tsetse, particularly subspecies of *G. palpalis* in west Africa and *G. fuscipes* in Central Africa and the Lake Victoria basin and upper reaches of the river Nile in East Africa. In East and Southern Africa, the disease is caused by *T. b. rhodesiense* (Rhodesian HAT, r-HAT) transmitted largely by savanna tsetse such as *G. morsitans* and *G. pallidipes*. Both forms of the disease are generally fatal if untreated.

WHO aims to eliminate HAT as a public health problem by 2020 and interrupt all transmission by 2030. The 2020 goal includes the aims of reducing global incidence of HAT to 'fewer than 2000 cases reported globally' and 'gambiense HAT incidence to less than 1 new case per 10,000 population at risk, in at least 90% of foci' [1]. In the period 1990–2018, there were 455,086 reported cases of HAT; most (440,055/455,086 = 96.7%) were cases of Gambian HAT and of these, two-thirds (301,819/440,055 = 68.6%) were in the Democratic Republic of the Congo (DRC) [2]. During this period, global incidence of g-HAT has declined from a peak of 37,385 cases/year in 1998 to 953 in 2018, the lowest number of HAT cases reported in the history of g-HAT surveillance. The annual incidence in DRC shows a similar trend with cases declining from 26,318 to 660 in the period 1998–2018. The decline globally and in DRC is principally due to large-scale active screening and treatment of the human population [3].

The rapid and sustained decline in the number of cases reported globally means that the 2020 goal of <2,000 cases being reported annually was achieved in 2016 (1,768 cases reported) and has continued to decline year on year. However, there will still be some important foci

where HAT is predicted to persist for decades. Analyses of interventions conducted in parts of the former Provinces of Bandundu and Equateur in DRC [4,5] suggest that active screening and treatment will not reduce incidence to <1 case/10,000 people in some Health Zones before 2020. Their analyses suggested however that the elimination goals could be achieved in these Health Zones by adding vector control to the current practice of actively screening the human population [6].

Efforts to control populations of riverine tsetse which transmit *T. b. gambiense* has not formed an important part of efforts against g-HAT. Vector control was not widely used in DRC because the methods available were prohibitively expensive, difficult to apply and/or ineffective. In the past decade however, two simple and cost-effective methods effective against riverine vectors of g-HAT have been developed. First, in areas where cattle are present in sufficient densities (>10 animals/km$^2$), tsetse can be controlled by treating cattle with pyrethroids [7,8]. Second, where cattle are absent or sparse (<10 animals/km$^2$), pyrethroid-treated targets, which simulate mammal hosts, can be deployed along the banks of rivers and other water bodies where tsetse concentrate [9–11]. Tsetse are attracted to them and then contact the cattle or targets and in so doing pick up a lethal dose of insecticide. Imposing a modest daily mortality of ≥4% on the female population can eliminate isolated populations of tsetse [12], and reduce densities of non-isolated populations by ~90% [9].

In the past decade, 'Tiny Targets', have been successfully used to control tsetse in g-HAT foci in Uganda [9], Guinea [10] and Chad [11]. Tiny Targets comprise small (25 cm high x 50 cm wide) panels of blue polyester with a flanking panel of black polyethylene netting, both impregnated with deltamethrin insecticide. Tsetse are attracted to the blue panel and as they approach the target they collide with the netting, picking up a lethal dose of insecticide. We hypothesized that Tiny Targets could also be used to accelerate the elimination of persistent foci of g-HAT in DRC.

The most important vector of g-HAT in western parts of DRC, including Kwilu Province (part of the former Province of Bandundu), is *G. f. quanzensis*. Behavioural studies have shown that this species is responsive to Tiny Targets [13]. Accordingly, we implemented a large-scale trial of Tiny Targets in Yasa Bonga Health Zone, one of the foci where the addition of vector control to screening activities is predicted to accelerate progress towards the elimination goal [4,6]. A series of large-scale entomological surveys of *G. f. quanzensis* in Yasa Bonga and its neighbouring health zones of Masi Manimba and Mosango (Fig 1) were conducted to capture information on the distribution of tsetse throughout the area both prior to and following the deployment of targets. and used remotely-sensed data to produce a map of potential tsetse habitat for this important vector, which guided the deployment of targets and allowed us to assess their impact. The objectives of this study were to (a) develop tsetse habitat suitability maps of the area using the pre-intervention entomological surveys combined with remotely sensed data, and (b) combine these maps with data from the subsequent post-intervention (monitoring) surveys to measure the impact of the intervention.

## Methods

### Study area

Studies were conducted between January 2015 and December 2017 primarily in the Yasa Bonga Health Zone (surface area≈2,800 km$^2$) of Kwilu Province, DRC, with a small number being conducted in the neighbouring Masi Manimba and Mosango health zones (Fig 1). The human population of Yasa Bonga is estimated to be ~180,000 people [14]. The Yasa Bonga Health Zone is a largely rural area, the population is engaged in cultivation, particularly cassava, and livestock are only present in low numbers. The Health Zone includes three large

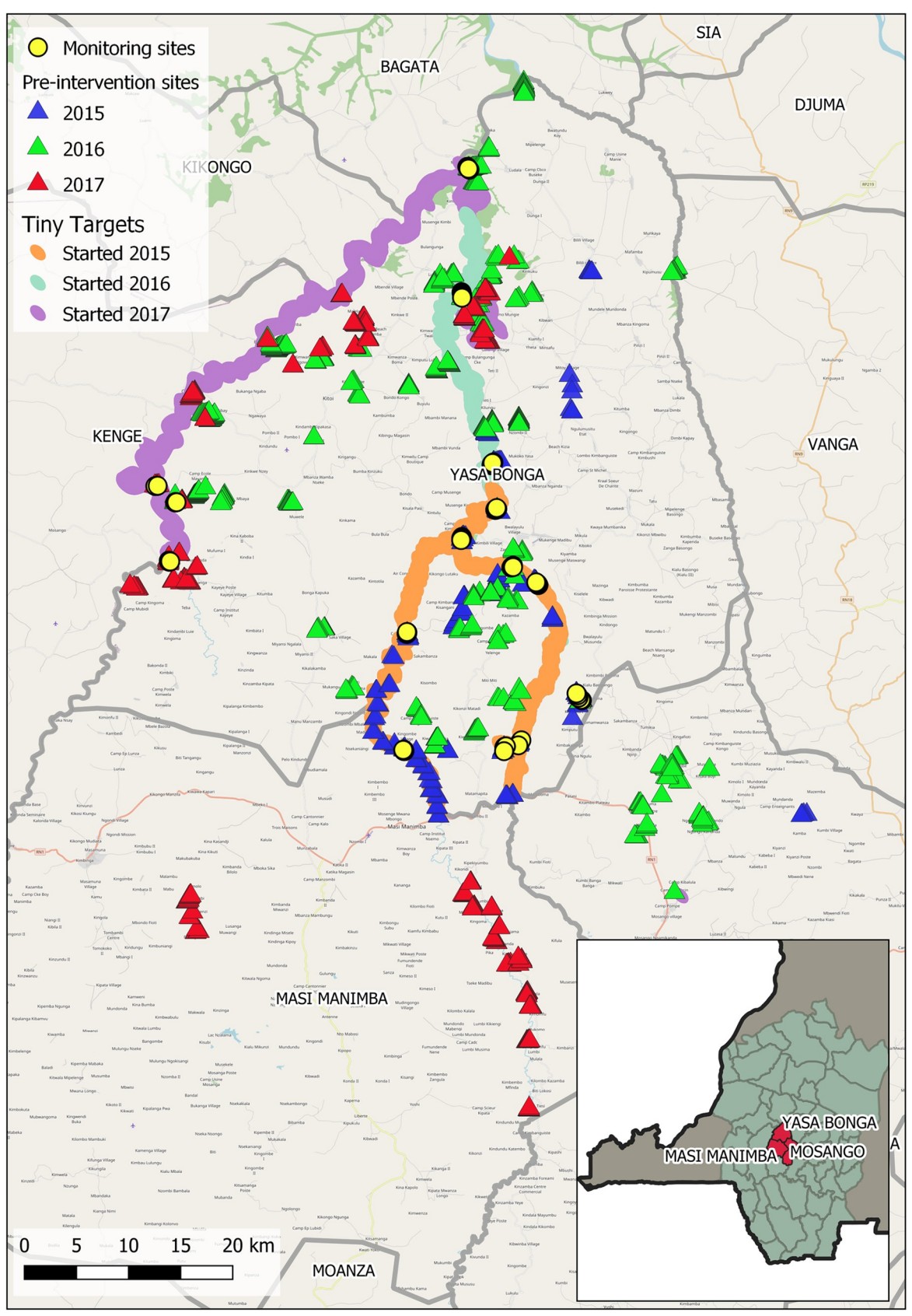

**Fig 1. Locations of the 1,234 tsetse monitoring sites at which tsetse flies were collected between January 2015 –December 2017.** Pre-intervention sites i.e. sites where data were collected prior to or unaffected by the intervention are triangles colour-coded by year of collection. Monitoring sites i.e. sites sampled immediately prior to the intervention, then repeatedly at regular intervals following the intervention are prepresented by circles. The river networks across which Tiny Targets have been deployed is also presented, colour-coded colour-coded according to the year in which the intervention was first introduced.

rivers (Lukula, Kafi, Inzia) and many of the tributaries of these rivers have been altered to form fish ponds. Along the banks of the water courses are bands of dense riverine vegetation including *Dalbergia glandulosa*, *Julbernardia* spp, *Garcinia kola*, *Stereospermum kunthianthum*, which form the natural habitat for tsetse. Humans and reptiles are likely the main hosts across the Health Zone, with domestic pigs in some villages also being of importance. The main dry season is between June and August and there is a shorter dry period in January/early February.

## Pre-intervention surveys of tsetse

Prior to the deployment of Tiny Targets, pyramidal traps [15] were deployed at 1,234 individual locations (Fig 1) and revisited at 1–2 day intervals to collect captured tsetse. Data were collected over multiple field surveys conducted in 2015–2017, with the locations of these traps being driven by factors specific to each field visit i.e. exploratory surveys to determine whether flies could be found in specific environments (along stretches of river, around fishponds, in villages). Environmental features of interest were identified using remotely sensed data, whereas specific sampling locations within these environments were selected according to their accessibility, determined using local knowledge and maps of the area. The locations of all traps were recorded using a global positioning system (eTrex 20x, Garmin, Southampton, UK). These data including the corresponding coordinates are available in the supporting information (SI), S1 Table. In the first year of the study, vector control was first implemented in the southern part of Yasa Bonga on the Lukula and Kafi rivers and operations expanded northwards and westwards in subsequent years to include the Inzia river (Fig 1). Accordingly, entomological surveys were also focussed largely in the southern part of the Health Zone and then expanded ahead of the deployment of targets.

The deployment of Tiny Targets (Vestergaard, Lausanne, Switzerland) in Yasa Bonga was carried out by teams of local individuals who had no experience of tsetse control prior to the intervention. These teams were recruited and trained by LSTM and the Programme National de Lutte contre la Trypanosomiase Humaine Africaine (PNLTHA) on how to install a Tiny Target on the riverbank to ensure optimal performance and how to record the location of all Tiny Targets using GPSs. While in other countries that use Tiny Targets the deployment is typically done on foot, in Yasa Bonga the teams operated from traditional canoes, "pirogues", navigating down the rivers and deploying Tiny Targets on both riverbanks at 50-metre intervals. The teams deployed over relatively long distances of up to 40km of river within a one week period. They remained in the field during the deployment, camping along the rivers, which enabled them to cover these distances in a short time period. The targets themselves were also assembled locally with teams collecting and cutting wood to the appropriate length to be used as target supports and then gluing the targets and supports together.

Within this area, Tiny Targets were deployed at 6 month intervals because the performance of the insecticide declines after this period [9]. Within the study period, Tiny Targets were first deployed in July-August 2015 (4,782 targets deployed) and thereafter in January-February and July-September of 2016 (12,421 targets deployed) and 2017 (18,883 targets deployed) (Fig 1).

### Post-intervention longitudinal monitoring of tsetse

Based on results from the pre-intervention surveys of tsetse, we selected 23 locations which produced the largest tsetse catches to monitor the impact of targets. At these locations, groups of up to eight pyramidal traps [15] were deployed to monitor changes in the catch of tsetse following the deployment of targets. Traps were deployed, each >100 m from its nearest neighbour, for 48 h and flies collected, identified and then stored in 100% ethanol for subsequent laboratory-based analyses. These sentinel surveys were conducted at three monthly intervals following the initial Tiny Target deployment.

### Environmental data

Environmental data commonly related to tsetse habitat were extracted from publicly available remote sensing sources including Sentinel-2 (10m resolution), Landsat 8 (30m+ resolution) and the Shuttle Radar Topography Mission (SRTM, 30m resolution). Sentinel-2 and Landsat 8 images were initially sought for both the main dry season (June–August) and the longer wet season corresponding to the time periods over which the exploratory traps were deployed. Due to limitations such as cloud cover, no suitable Sentinel-2 wet season images were available for the targeted time period and a single dry season image was identified for July 2016. Landsat 8 images were available for both the wet (December 2014) and dry (August 2015) seasons. Due to these limitations in our ability to obtain regular images it was not possible to explore the impact of interannual variability in temporally varying environmental factors in this study.

Environmental data extracted from these remotely sensed data sources included landcover class (water, forest, shrubland, grassland, bare earth), obtained by applying a supervised classification algorithm using the QGIS Semi-Automatic Classification Plugin (SCP) [16] to the Sentinel-2 imagery, Enhanced Vegetation Index (EVI) [17] and land surface temperature (LST) which was obtained using the Landsat 8 thermal band and the QGIS LST plugin [18]. A river network was derived by applying hydrological tools to the SRTM digital elevation model (DEM) data to calculate flow accumulation, and a pixel was determined to be a river if the flow accumulation exceeded a value of 20,000. A full list of derived environmental data is contained in S2 Table.

### Exploratory trap data analysis

Summaries for the trap data, including both the pre-intervention traps and the subsequent monitoring traps, stratified by season (wet or dry) and whether they were likely to be impacted by the intervention were produced. Impact status was determined by the distance between the trap and any recently deployed targets i.e. traps were determined to be either in the intervention area (<500m from a recently deployed target), at the edges of the intervention area (500m-5km from a recently deployed target) or uninfluenced by targets (pre-intervention or more than 5km away from a recently deployed target). An initial assessment of the effect of season on fly numbers was made by fitting a simple negative binomial generalised linear mixed model (GLMM) to the count data with season as the only fixed effect, and trap ID as a random effect to account for repeated visits to a single trap. Similarly, a negative binomial GLMM was fitted to the count data with intervention status as the only fixed effect. All models were fitted in R (v 3.6.1) using the lme4 package [19].

### Habitat suitability mapping

Data collected during the shorter dry season were used to produce a habitat suitability map. To remove the influence of Tiny Targets on the fly population, the habitat suitability analysis

was undertaken using both (a) data from traps that were deployed prior to the intervention and (b) traps that were located at least 5km from the nearest recently deployed target (where 'recent' is defined as within the last 8 months). These thresholds were considered conservative enough that the intervention was unlikely to be influencing the fly population. If a site was visited multiple times during a season, regardless of year, then it was considered positive if at least one fly was caught during any visit. The total number of days over which the trap was observed was also recorded.

Two methods of producing dry-season habitat suitability maps were considered, namely boosted regression trees (BRT) and regularised logistic regression. A BRT approach makes use of two algorithms. Regression trees are a classification approach that recursively partitions the data using a series of binary rules ('split points') applied to the predictor variables, culminating in the observation being assigned to one of the specified classes [20,21]. With each decision rule the tree 'grows', stopping only once particular stopping criteria are met e.g. once the predictive performance of the tree reaches an acceptable level. Boosting is a method of improving model accuracy. In relation to regression trees, the boosting technique is an iterative process in which an initial regression tree is fitted (Tree 1), an additional tree is fitted to the residuals of the model. The two trees are then combined (Tree 2) and a regression tree is fitted to the residuals of this new model. This new tree is combined with Tree 2 to form Tree 3. The process is repeated many times, and the resulting trees are then combined to form the final BRT model [20].

Regularised regression is a form of regression which is designed to reduce overfitting by allowing the coefficient estimates to shrink towards zero. This is undertaken by penalising the size of the coefficients while simultaneously minimising the difference between the observed and resulting fitted values. Commonly used regularisation techniques include LASSO (Least Absolute Shrinkage and Selection Operator) regression (which used L1 regularisation) and ridge regression (which used L2 regularisation). L1 regularisation derives a penalty based on the absolute value of the size of the coefficient whereas L2 regularisation derives a penalty based on the square of the size of the coefficient [22–24]. LASSO regression can be considered as a method of model selection as it results in coefficient estimates of 'unimportant' covariates of zero, thus effectively removing these covariates from the model. In this paper, LASSO regression is applied to select which environmental variables are associated with tsetse presence/absence and subsequently derive the habitat suitability maps.

A total of 15 environmental covariates derived from the remotely sensed data presented in Table 1 were considered. These are listed in the SI (S2 Table) and include remotely sensed data extracted at the location of the traps, plus summaries of these data within a 350m buffer of the traps. Experimental estimates of the daily movement of tsetse range between 100-800m/day [25] with subspecies of *Glossina fuscipes* being in the range of ~350m/day [26].

Due to the small number of positive sites, it was not suitable to separate the data into training and testing sets. Therefore the caret package in R [27] was used to fit the models (including selecting the tuning parameters for the selected model fitting approach) to the entire dry

**Table 1. Sources of environmental risk factors under consideration.**

| Source | Environmental factors | Time | Spatial resolution |
| --- | --- | --- | --- |
| SRTM DEM | Elevation, slope, river network derived using a flow accumulation threshold of 20,000 | February 2000 | 30m |
| Sentinel 2 | Land cover classification, enhanced vegetation index | 14th July 2016 | 10m |
| Landsat 8 | Land surface temperature | Day 233 (August) 2015 and Day 342 (December) 2014 | 100m, resampled to 30m |

season dataset using a repeated 5-fold cross-validation approach i.e. for each combination of tuning parameters, the data were split into 5 subsets, the model was fitted to 4 of these subsets, and then predictions were made for the excluded data. The fit of each of these models was assessed using the Brier score as the performance metric. The Brier score measures the difference between the observed data and the fitted values which were 1 if tsetse were observed/predicted to be present, and 0 otherwise. More specifically, the Brier score is defined as $\frac{1}{n}\sum_{i=1}^{n}(y_i - \hat{p}_i)^2$ where $y_i = 1$ if flies were present at location $i$, and zero otherwise, and $\hat{p}_i$ is the cross-validated predicted probability that flies were present. As such, the Brier score lies between 0 (perfect predictions) and 1 (all predictions are incorrect).

The entire process was repeated 100 times in total to reduce the bias introduced through the partitioning of the data, and a summary of the 100 resulting Brier scores was produced. The optimal model was that which minimised the mean Brier score. Optimal values were selected for three tuning parameters (number of trees, learning rate and tree complexity) in the BRT model and one tuning parameter (lambda) for the LASSO regression. The performances of the two fitted models were compared both using the Brier score of the model fitted to the entire dataset, and the mean Brier score resulting from the cross-validation. In addition, the receiver-operator characteristic area under the curve (ROC-AUC) was also calculated which ranges between 0 and 100 percent, with 100% indicating perfect predictions. The ROC-AUC is a measure of how well the model can differentiate between positive and negative sites assuming a discriminatory probability threshold. The environmental covariates selected using both modelling approaches were also reported.

Maps of dry season habitat suitability expressed as a habitat suitability proportion were then produced for the entirety of Yasa Bonga and neighbouring Masi Manimba on a 10m pixel scale using the best performing modelling approach as determined by the fitted Brier score and ROC-AUC.

## Evaluation of the impact of Tiny Targets on tsetse distribution

The evaluation of the impact of the Tiny Targets on the distribution of tsetse was conducted using the Yasa Bonga sentinel sites i.e. 12 clusters of 8 traps along the Lukula and Kafi rivers of which 9 were surveyed 10 times and 3 were surveyed 8 times, and a further 10 clusters of 8 traps and one cluster of 6 traps along the Inzia river each of which had been surveyed twice. During this study period, targets were deployed 5 times within the initial intervention area (starting 2015), 3 times in the northern expansion (starting 2016) and once around the Inzia river (starting 2017, Fig 1). Traps were determined to be either in the intervention area (<500m from a recently deployed target), at the edges of the intervention area (500m-5km from a recently deployed target) or uninfluenced by targets (pre-intervention or more than 5km away from a recently deployed target).

Initial exploratory analysis of the impact of targets on the tsetse distribution was undertaken through the calculation of summaries of flies per trap per day according to the distance between the traps and the targets, and the time between the catch and the target deployment. A series of negative binomial generalised linear mixed models were then fitted to the sentinel site count data. The models were developed to determine the influence of habitat suitability on fly counts, and the influence of the intervention after accounting for differences in habitat suitability between sites. Initially, the intervention status of the traps was defined using the distance categories specified above only ('in', 'edge', 'uninfluenced') following which the model was expanded to consider the number of previous interventions.

$$Y_{ijt} \sim NegBin(\mu_{ijt} = d_{ijt}\eta_{ijt}, \kappa)$$

$$\log\{\mu_{ijt}\} = \log\{d_{ijt}\} + \log\{\eta_{ijt}\}$$

$$= \beta_0 + \beta_k X_{ijkt} + U_j + V_{ij} + \log\{d_{ijt}\}$$

where $Y_{ijt}$ is the number of flies caught in trap $i$, cluster $j$ at observation time $t$, $\mu_{ijt}$ is the mean number of flies observed and $\eta_{ijt}$ is the rate of flies caught per trap per day. The coefficient $\beta_0$ represents the intercept, $X_{ijkt}$ are covariates associated with the trap observed at time t (proximity to intervention, habitat, season, time since the intervention was implemented, and relevant interactions) with $\beta_k$, k = 1,. . .,n representing the associated coefficients. The random effects in the model, $U_j$ and $V_{ij}$, represent the nested random effects, both of which are assumed to be normally distributed with mean zero and variance $\tau_u^2$ and $\tau_v^2$ respectively. The offset $d_{ijt}$ represents the number of days over which flies were caught at trap $i$, cluster $j$ at time $t$. Likelihood ratio tests were undertaken to compare nested models and evaluate the significance of both fixed and random effects using the lme4 package in R, and confirmed via parametric bootstrap using the package pbkrtest where appropriate [19,28]. In instances where the models being compared were non-nested, the Akaike Information Criterion (AIC) was used to select the best fitting model. A list of considered models and their corresponding evaluation metrics is contained within the SI (S2 Table).

Maps of the fitted random effects were then produced, and evidence of any spatial correlation in these was explored. Given the spatial sparsity of the sentinel sites, it was not feasible to formally check this, hence this primarily consisted of a visual inspection.

## Results

As of 13th September 2017 there were 2,474 trap data records at 1,234 locations (1,055 in Yasa Bonga, 110 in Masi Manimba and 69 in Mosango), resulting in 837 flies being captured over 4,952 trapping days. Table 2 summarises fly catches from these traps according to season and

**Table 2. Summary of recorded tsetse data by intervention status and season.** Intervention status consists of three categories: (1) traps are considered unaffected by the intervention if they were deployed prior to the intervention or were placed more than 5km away from recently deployed targets, (2) traps were placed 500m-5km from a recently deployed target, (3) traps were placed <500m from a recently deployed target. Data from 2015–2017 are pooled by wet (September–May) and dry (June-August) season.

|  |  | Wet season (Sep–May) | Dry season (June-August) | Total |
|---|---|---|---|---|
| **Unaffected by the intervention** | No of records | 777 | 360 | 1,137 |
|  | No of trapping days | 1,536 | 750 | 2,286 |
|  | No of flies caught | 248 | 248 | 496 |
|  | Flies/trap/day | 0.16 | 0.33 | 0.22 |
| **Between 500m-5km from the intervention** | No of records | 387 | 180 | 567 |
|  | No of trapping days | 807 | 328 | 1,135 |
|  | No of flies caught | 107 | 140 | 247 |
|  | Flies/trap/day | 0.13 | 0.43 | 0.22 |
| **Within 500m of the intervention** | No of records | 516 | 254 | 770 |
|  | No of trapping days | 1,033 | 498 | 1,531 |
|  | No of flies caught | 46 | 48 | 94 |
|  | Flies/trap/day | 0.04 | 0.10 | 0.06 |
| **All traps** | No of records | 1,680 | 794 | 2,474 |
|  | No of trapping days | 3,376 | 1,576 | 4,952 |
|  | No of flies caught | 401 | 436 | 837 |
|  | Flies/trap/day | 0.12 | 0.28 | 0.17 |

intervention status ($\leq$ 500m, 500m-5km, or >5km/pre-intervention[unaffected] from a recently deployed target) in addition to the number of observation days and total number of flies caught. Across the 1,234 individual trapping sites a total of 837 flies were caught over 4,952 trapping days (0.17 flies/trap/day). In comparing data between the wet and the dry season, the flies/trap/day over the nine-month wet season (0.12 flies/trap/day) was lower than that of the shorter dry season (0.28 flies/trap/day) with this difference being significant (p<0.0001) as demonstrated by fitting a negative binomial GLMM to the data with season as a fixed effect and trap ID as a random effect.

## Habitat suitability maps

Table 3 presents a summary of available pre-intervention tsetse count data by year and season. Dry season data were used to develop the habitat suitability maps i.e. data from 332 locations of which 20% (65/332) caught at least one fly during the observation period. The observation period ranged between one and four days, with the majority of locations being observed for two days (63%, 208/332).

The relative importance of each of the environmental variables with respect to their contribution to the final BRT and LASSO models is presented in S2 Table of the SI. In both models the percentage of forest cover within a 350m radius, the range of EVI within a 350m radius, the mean topographical water index (TWI) within a 350m radius and land surface temperature are important contributors to the model. Maximum TWI is also identified as important to the fit in the BRT model, whereas ruggedness is selected for the LASSO model. Optimal values of the model parameters based on minimising the Brier score for the BRT model were number of trees = 150, interaction depth = 5, shrinkage = 0.01. The optimal value of lambda in the LASSO model was 0.003.

The evaluation measures for both the resulting BRT and LASSO models fitted using the 332 dry season traps and their associated environmental covariates are presented in Table 4. In comparing the performance of the two models, the BRT summaries are consistently more optimal than those obtained using LASSO when considering the full model fit e.g. the Brier score for the fitted model is 0.0975 for the BRT model in comparison to 0.1276 for the LASSO model. With regards to the cross-validated summaries, the BRT model is the better performing model when considering ROC-AUC, but is marginally worse when considering the Brier score.

A map of the estimated distribution of tsetse obtained using the optimal BRT model is presented in Fig 2. The habitat is estimated to be more suitable for tsetse flies along the river running down the centre of the Yasa Bonga Health Zone (Lukala), with habitat along the other main rivers in Yasa Bonga and Masi Manimba being less suitable.

**Table 3. Summary of the number of pre-intervention trap deployments by year and season (wet/dry).** Date from these sites were used to develop the habitat suitability maps. Unique locations refers to the number of locations where at least one trap was deployed over the three year period. Positive sites are those where at least one fly was captured during any deployment.

|  | Wet season (September–May) | Dry season (June-August) | Total |
|---|---|---|---|
| **2015** | 315 | 88 | 403 |
| **2016** | 292 | 177 | 469 |
| **2017** | 166 | 92 | 258 |
| **Total** | 773 | 357 | 1,130 |
| **Unique locations** | 710 (88 positive, 12%) | 332 (65 positive, 20%) |  |

**Table 4. ROC-AUC and Brier score of the final fitted BRT and LASSO models plus the median andrange of the ROC-AUC and Brier score for the 100 cross-validated predictions.**

| | | | Brier Score | ROC-AUC |
|---|---|---|---|---|
| Final model | BRT | | 0.0975 | 0.9402 |
| | LASSO | | 0.1276 | 0.8089 |
| Cross-validation | BRT | Median | 0.1410 | 0.7583 |
| | | Range | 0.1336–0.1491 | 0.7207–0.7843 |
| | LASSO | Median | 0.1393 | 0.7491 |
| | | Range | 0.1356–0.1575 | 0.7071–0.7727 |

## Evaluation of the impact of Tiny Targets on tsetse distribution

In comparing the number of flies caught in pre-intervention traps (i.e. traps either deployed prior to the intervention, or those more than 5km away from a recently deployed target) and post-intervention traps (those within 0-500m of a recently deployed target) the number of flies per trap per day reduces from 0.22 to 0.06 (Table 2). In the GLMM fitted to the fly count data from all traps, with an intervention factor (0-500m, 500m-5km, >5km from the intervention) and trap site-level random effects only, this intervention effect is significant. Within the wet season the rate reduces from 0.16 to 0.04 (within 500m of the intervention) whereas in the dry season the rate reduces from 0.33 to 0.10 flies/trap/day. In considering both the effect of season and the intervention on fly rate simultaneously, there does not appear to be any interaction between the two i.e. the effect of the intervention on the fly rate is the same both in the wet and dry season.

In addition to considering the intervention by three distance categories, the effect of target distance as a more continuous measure and time since target deployment were also explored. When considering the effect of target distance, it was observed that initially as the distance increased, both the proportion of traps with flies present (Fig 3) and flies per trap per day increases, then declines when the distance exceeds 5km. This decline is likely as the traps being placed far from targets are in areas of less suitable habitat (e.g. far from rivers and streams) than those closer to targets. Summaries of the estimated dry season habitat suitability by distance to the nearest target for all traps is presented in Table 5, indicating a decline in suitable habitat at distances greater than 500m from a target.

Time since target deployment on the tsetse population was stratified into three categories: 0–2 months, 3–5 months, >6 months/never). Table 6 summarises the observed fly data by the time category and season. No clear pattern between time since the target was deployed and the fly catch data was observed. A simple negative binomial GLMM with time since deployment as the only fixed effect suggests no difference between catches made 0–2 month after target deployment and 3–5 months after target deployment (p = 0.1170), suggesting that the impact of the targets is rapid and persistent. No interaction was found between the time since deployment and season.

Sentinel site data only were then used to explore the effect of the intervention over time. For simplicity, sentinel traps that changed intervention status between the initial pre-intervention survey (T0) and the last recorded sentinel survey as a result of the expansion of the intervention were excluded from the analysis. This left 889 traps from 20 clusters which were observed between two and nine times over the study period. Prior to the implementation of the intervention the mean catches for these traps was 0.7 flies/trap/day, ranging from 0 to 19.5 flies/trap/day. An exploratory analysis of the temporal structure of the data (Fig 4) indicated that there did not appear to be a continuous cumulative effect of the intervention, with the number of flies in the traps placed within 500m of a target being consistently low following the

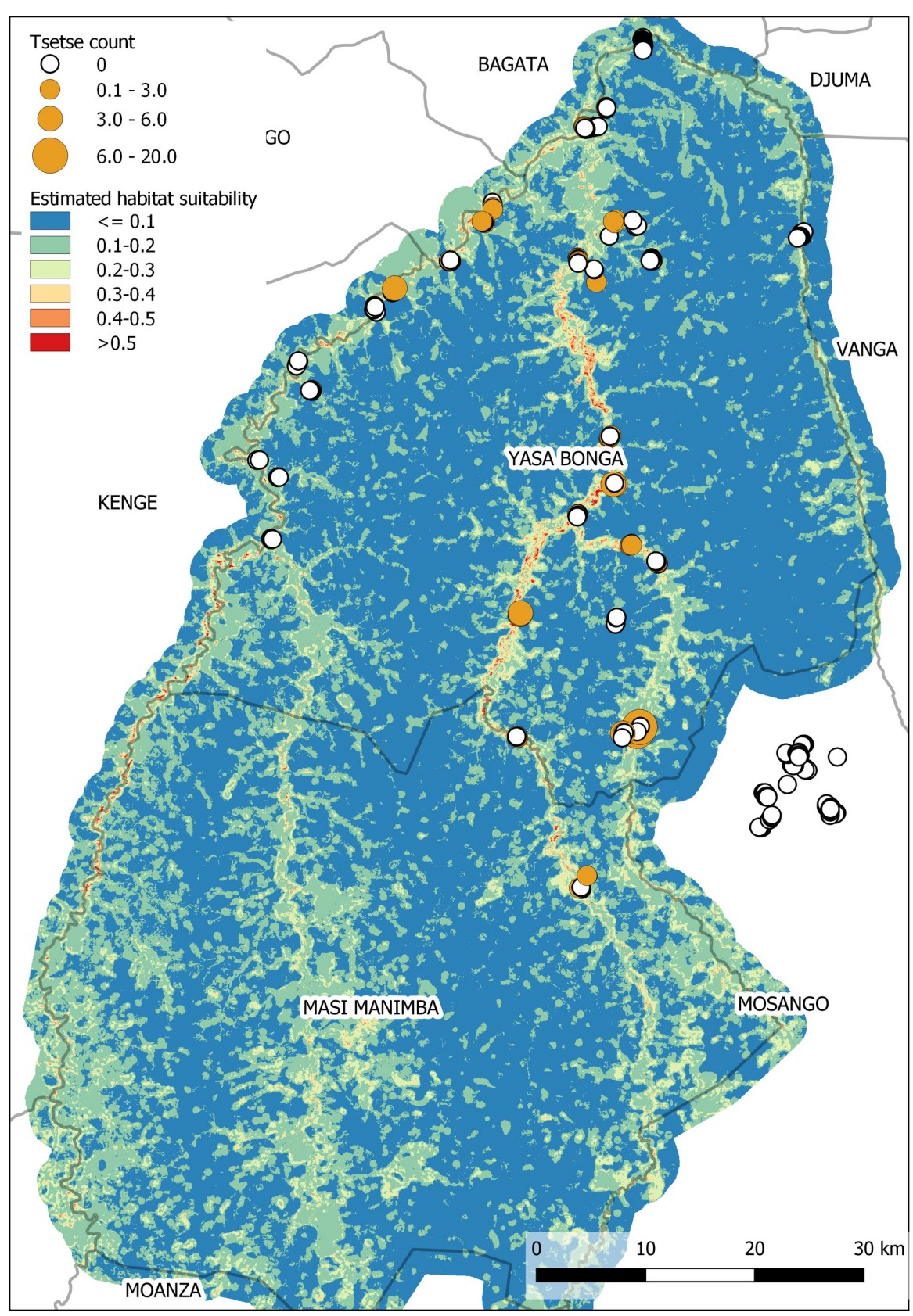

**Fig 2. A map of habitat suitability obtained using a boosted regression tree approach.** The fly counts (flies/trap/day) of each of the 332 trap sites used to fit the model is also displayed.

initial implementation of the intervention which occurred between observations made at T0 (0.78 flies/trap/day, 95% CI 0.676–0.900) and T1 (0.11 flies/trap/day, 95% CI 0.070–0.153) i.e. <20% of the value observed at baseline. As a result, the effect of time in the subsequent model was included as either a binary variable (pre-intervention, post-intervention) or a three level factor (pre-intervention, within 6 months of the initial implementation [T1-T2]), >6 months after the initial intervention [T3 onwards]). Similarly, the effect of targets based on the distance from the intervention was considered as a binary variable (in the intervention area, unaffected) using a distance threshold of 500m, and a three level factor (in, edge, unaffected) as described previously.

Covariates considered in the GLMM were therefore the proximity of the intervention (two-level or three-level factor), time since the intervention was initiated (two-level or three-level factor), habitat suitability derived from the BRT model and season (wet/dry). Both trap site and trap cluster were considered in the models as random effects. Table 7 displays the results of the final selected model. Intermediate model output is presented in S3 Table.

At baseline, there was no significant difference in the number of flies caught in traps set in areas with differing intervention status i.e. areas within the intervention area ('in', adjusted log-RR = 0.65, 95% CI = [-1.3275, 3.7609]), well outside of the intervention area ('out', reference level in the model) and intermediate areas ('edge', adjusted log-RR = 2.05, 95% CI =

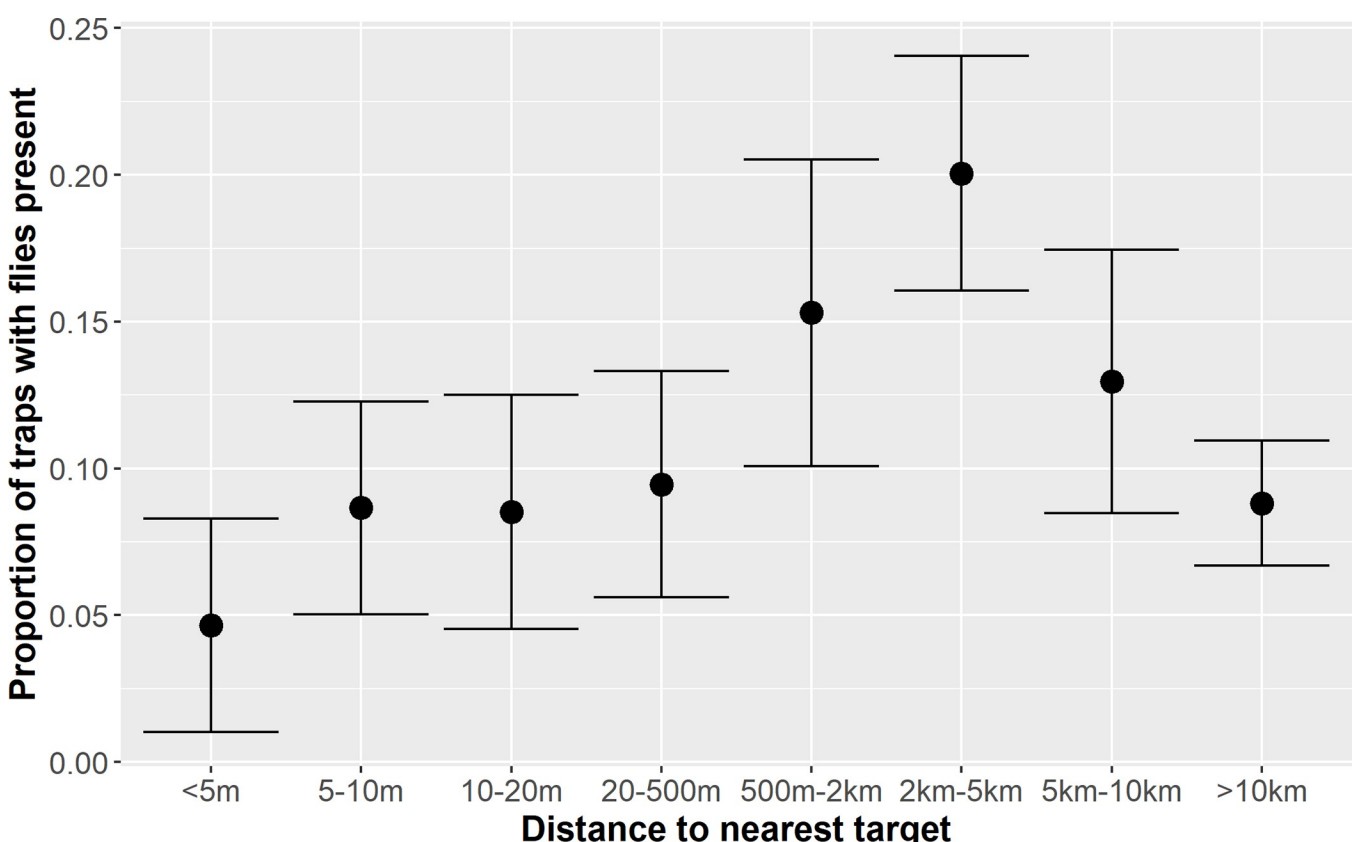

**Fig 3. Proportion and associated 95% confidence interval of traps in which flies were present by the distance to the nearest recently deployed target.**

**Table 5. Summaries of trap data (number of traps deployed and flies/trap/day) plus estimated dry season habitat suitability (median and interquartile range extracted from the BRT model) by distance to nearest Tiny Target.**

| | | 0-5m | 5-10m | 10-20m | 20-500m | 500-2km | 2-5km | 5-10km | >10km |
|---|---|---|---|---|---|---|---|---|---|
| **N traps** | | 129 | 231 | 188 | 222 | 183 | 384 | 216 | 681 |
| **Flies/trap/day** | | 0.05 | 0.12 | 0.13 | 0.16 | 0.30 | 0.50 | 0.28 | 0.23 |
| **Habitat suitability** | Median | 0.42 | 0.34 | 0.29 | 0.22 | 0.11 | 0.11 | 0.11 | 0.13 |
| | IQR | 0.28–0.50 | 0.23–0.49 | 0.21–0.45 | 0.14–0.38 | 0.09–0.17 | 0.07–0.16 | 0.08–0.20 | 0.09–0.23 |

[-0.9337, 4.4011]). As was observed in the exploratory analysis, there is a reduction in flies caught during the wet season (adjusted log-RR = -0.72, 95% CI = [-1.6445, 0.0523]).

Habitat suitability is significantly and positively associated with fly catches prior to the introduction of the intervention (adjusted log-RR = 10.17, 95% CI = [8.1608, 12.5348]), however this effect weakens once the intervention has been implemented (adjusted log-RR = 3.33, 95% CI = [1.0200, 5.9269]).

After accounting for the effect of habitat, once the intervention has been implemented (observation period T1 onwards), the impact on the number of flies caught varies significantly according to the location of the trap and the duration of the intervention. Within 6 months of the implementation of the intervention (monitoring operation T1 and T2) a reduction in fly counts is observed in traps set within 500m from a target, and in the 'edge' areas (within 500m: adjusted log-RR = -3.25, 95% CI = [-4.5682, -1.5294]; 'edge' areas: adjusted log-RR = -5.90, 95% CI = [-9.8315, -2.5211]). The reduction in subsequent monitoring periods (>6 months, T3 onwards) continues in the immediate intervention area whereas there is a slight increase in risk of flies in edge areas in comparison to the previous time period (within 500m: adjusted log-RR = -2.97, 95% CI = [-4.6055, -1.1405]; 'edge' areas: adjusted log-RR = -3.12, 95% CI = [-6.0659, -0.3326]), indicating there may be some recovery in the fly population, due to either local rebound or reinvasion of tsetse from neighbouring areas, despite continued deployment of the intervention.

To illustrate this further, the estimated number of flies per trap per day obtained from the fitted model under a range of scenarios and their corresponding simulation-based confidence intervals have been calculated from the fitted model, assuming a habitat suitability equal to the mean of the sentinel sites (0.311), dry season collections and a trapping period of two days (Table 8).

Likelihood ratio tests conducted on models with cluster-level random effects only and both cluster and site-level random effects indicated both effects were present in the data (p = 0.0001). A map of the conditional modes of the cluster-level random effects (S1 Fig)

**Table 6. Summary of fly data (number of traps deployed, number and percentage of traps where tsetse were present and flies/trap/day) by the number of months since the nearest target was deployed (0–2, 3–5, 6+ or pre-intervention/never) and season (wet/dry).**

| | Months since target deployment | N | Tsetse present | % of traps with tsetse present | Flies/trap/day |
|---|---|---|---|---|---|
| **All traps** | 0–2 | 1229 | 138 | 11.2 | 0.12 |
| | 3–5 | 678 | 74 | 10.9 | 0.11 |
| | 6+/Never | 567 | 119 | 21.0 | 0.34 |
| **Dry season** | 0–2 | 267 | 46 | 17.2 | 0.23 |
| | 3–5 | 307 | 43 | 14.0 | 0.18 |
| | 6+/Never | 220 | 59 | 26.8 | 0.46 |
| **Wet season** | 0–2 | 962 | 92 | 9.6 | 0.09 |
| | 3–5 | 371 | 31 | 8.4 | 0.07 |
| | 6+/Never | 347 | 60 | 17.3 | 0.26 |

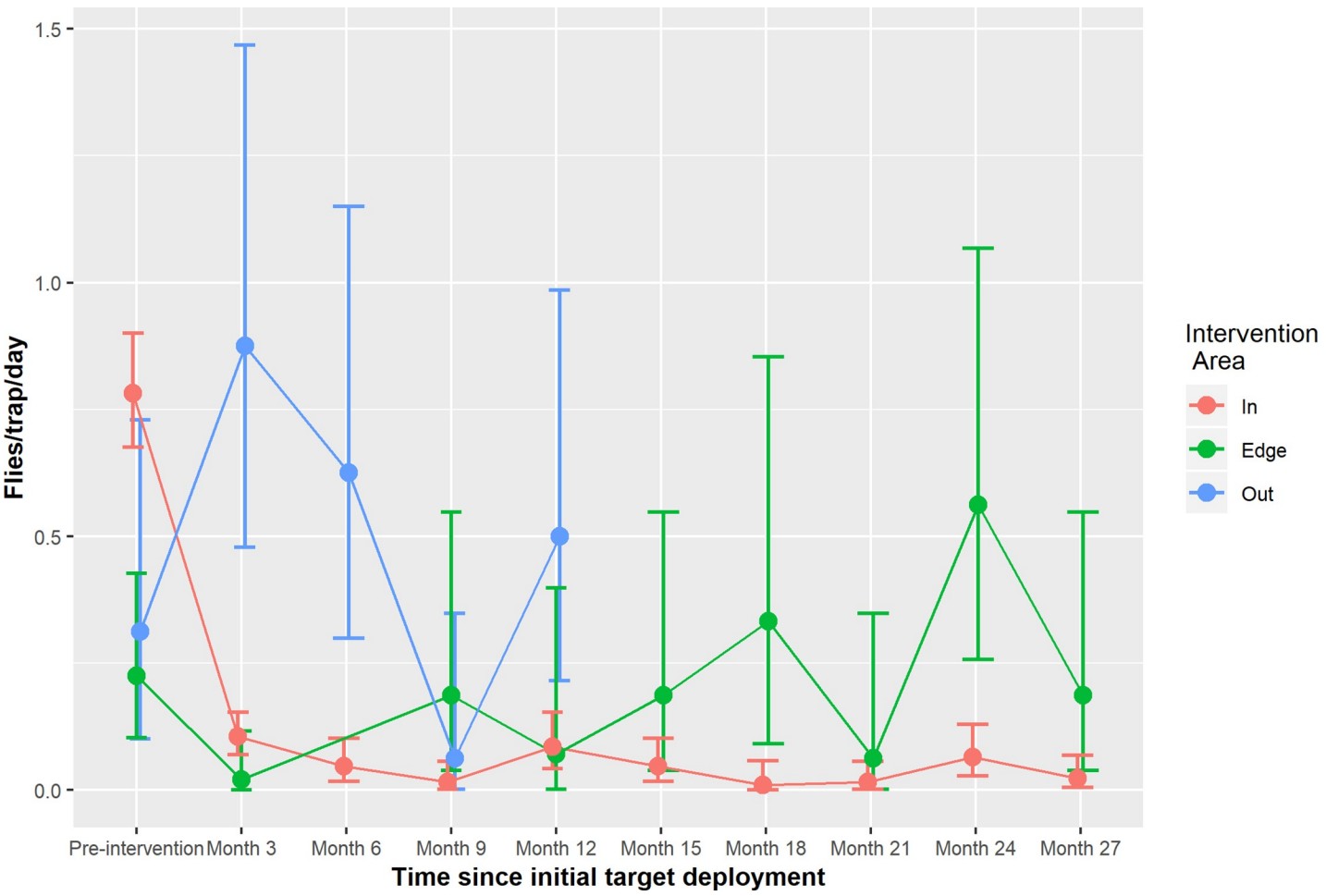

**Fig 4. Flies per catch per day by observation period and intervention area, and 95% confidence intervals.**

indicates no obvious spatial pattern, however more spatially diverse data are needed to explore this more formally.

## Discussion

### Habitat suitability maps

Two modelling approaches were considered to derive a map of habitat suitability for the intervention health zones. Both approaches allow complex relationships to be accounted for between tsetse abundance and the surrounding environment and performed similarly with respect to being able to predict where tsetse were expected to be found. Further, both approaches identified a similar set of environmental factors most strongly associated with tsetse habitat, namely average topographical wetness index, percentage of the surrounding area classified as forest, the range of enhanced vegetation index all within a 350m radius of a trap, plus the estimated land surface temperature. This information and the resulting suitability map can be used by tsetse control programmes to validate their initial choices of where to deploy targets and identify additional areas in which to either expand the intervention or to establish additional monitoring sites. Further research is required to determine the link between suitable tsetse habitat and the geographical distribution of reported g-HAT cases.

**Table 7. Model output (adjusted log-relative risk and corresponding 95% confidence intervals) for each component of the final selected GLMM fitted to the sentinel site data.**

| | | Adjusted log-relative risk | 95% confidence interval |
|---|---|---|---|
| Intervention status at baseline | Out | - | - |
| | Edge | 2.0468 | -0.9337, 4.4011 |
| | In | 0.6487 | -1.3275, 3.7609 |
| Season | Dry | - | - |
| | Wet | -0.7229 | -1.6445, 0.0523 |
| Intervention duration | Baseline | - | |
| | Up to 6 months | 4.3039 | 2.4340, 6.0164 |
| | Over 6 months | 2.7054 | 0.8050, 4.5523 |
| Habitat suitability | Pre-intervention | 10.1749 | 8.1608, 12.5348 |
| | Post-intervention | 3.3312 | 1.0200, 5.9269 |
| Intervention status post-intervention | Up to 6 months, intervention area (<500m) | -3.2549 | -4.5682, -1.5294 |
| | Over 6 months, intervention area (<500m) | -2.9710 | -4.6055, -1.1405 |
| | Up to 6 months, edge areas | -5.9018 | -9.8315, -2.5211 |
| | Over 6 months, edge areas | -3.1206 | -6.0659, -0.3326 |

These forms of habitat suitability models assume that traps which do not catch any flies are in areas less suitable for tsetse than those traps where flies are found. A weakness in this approach is that many factors other than the surrounding habitat may explain the observed absence of flies. As many traps were observed at multiple time points during the dry season study period, it was possible to partially account for this by assuming that flies were present if they were observed at least once during any trapping period. Further, the habitat suitability models and resulting maps were developed for the short dry season only. It was not possible to obtain contemporary remotely sensed imagery for the entire trapping period, hence an assumption was made that environmental data obtained for a single time point adequately represented the dry season across multiple years. It was not possible to obtain sufficiently cloud-free imagery for the wet season from the Sentinel-2 satellite and as such no wet season habitat suitability models were derived. It was noted that fly abundance was generally lower in the wet season compared to the dry season for reasons not well understood. A number of potential factors may contribute to the reduction. First, the area of suitable habitat may expand during the wet season leading to an overall reduction in density as flies disperse. Second, the saturated soil may be less suitable for puparia. Third, localised and transient reductions in the temperature associated with rainfall may reduce tsetse activity and hence the numbers attracted to traps. Further investigations are required to test these hypotheses and generate new ones.

In this study we found that there were modest seasonal variations in catches of tsetse with a mean of 0.04 tsetse/trap/day in the wet season (Sep-May) compared to 0.1 tsetse/trap/day in the dry season (June-August). De Deken et al. (2005) [29] also reported that catches of *G. f.*

**Table 8. Estimated flies/trap/per day under a range of intervention scenarios obtained from the final fitted model.**

| | Pre-intervention | | T1-T2 (up to 6 months post intervention) | | T3 onwards (>6 months post intervention | |
|---|---|---|---|---|---|---|
| | Estimate | 95% CI | Estimate | 95% CI | Estimate | 95% CI |
| **Within 500m** | 0.0609 | 0.0077, 0.4968 | 0.0202 | 0.0022, 0.1756 | 0.0054 | 0.0007, 0.0440 |
| **Edge** | 0.2346 | 0.0188, 2.8900 | 0.0059 | 0.0002, 0.1538 | 0.0193 | 0.0013, 0.2526 |
| **Unaffected** | 0.0308 | 0.0013, 0.7278 | 0.2646 | 0.0134, 5.5911 | 0.0540 | 0.0025, 1.2276 |

*quanzensis* at sites in the Kisenso District of DRC were generally low with numbers being greatest in December-January (~0.7 tsetse/trap/day) during the wet season (defined as October-April in their study) and lowest (~0.5 tsetse/trap/day) at the start of the dry season (May to July). Tirados et al. (2015) [9] also found that seasonal fluctuations in the catches of *G. f. fuscipes* in Uganda were relatively small (0.7–3.9 tsetse/trap/day) and studies of the population genetics of *G. f. fuscipes* in Uganda also suggest that populations are temporally stable [30–32]. This temporal stability in populations of G. f. fuscipes contrasts with the strong seasonal fluctuations displayed by savanna species such as *G. morsitans morsitans* and *G. pallidipes* where there are >10-fold differences in catches between seasons [33]. The stability of *G. fuscipes* populations is probably related to the more equable seasons found in central Africa where hot dry seasons are shorter and less extreme.

## Evaluation of the impact of Tiny Targets on tsetse distribution

The catches of tsetse prior to the deployment of Tiny Targets were generally low relative to other areas where sleeping sickness occurs and *G. fuscipes* is the main vector. For instance, in the West Nile focus of Uganda, daily catches of *G. f. fuscipes* were ~2.7 flies/trap (ranging from 0–144) [9] compared to 0.7 tsetse/trap (ranging from 0–19.5) in the present study. This low apparent density is unexpected given that the high incidence of g-HAT in Yasa Bonga [4]. The low catches of *G. f. quanzensis* in the current study may reflect differences in the sampling efficiency of pyramidal traps for *G. f. quanzensis* and *G. f. fuscipes*. There are no data to allow us to estimate any such difference but studies of other species of tsetse show that there can be large inter-specific differences in sampling efficiency [34]. Alternatively, *G. f. quanzensis* may be a more effective vector of *T. b. gambiensis* by, for example, taking a higher proportion of blood-meals from humans. While the abundance of tsetse is an important factor in disease risk, transmission depends on, amongst other things, abundance of hosts at points where humans and/or animal hosts and tsetse contact each other. Our study did not include fine scale analyses of the abundance and movement of humans and animals and such work might explain why incidence of g-HAT is high in places where catches of tsetse are low.

Nonetheless, we were still able to observe that Tiny Targets have a large effect on the fly population in areas close to the intervention (<500m) such that within three months of the targets being deployed catches decreased by >85% compared to their pre-intervention level and remained so throughout the monitoring period. Epidemiological models of g-HAT in Yasa Bonga [5] suggest that interruption of transmission can be achieved by reducing the abundance of tsetse by > 60%. Our findings compare to similar declines for *G. f. fuscipes* in Uganda but is not as marked as that for *G. f. fuscipes* in Chad [11]. The effect diminishes with distance although some benefits are seen in traps situated 500m-5km from targets. This distance effect could not however be fully investigated due to the distribution of traps with respect to distance from targets as few traps were placed far from targets, hence further investigation is needed.

In Yasa Bonga, large numbers (>9,000/deployment) of Tiny Targets were deployed rapidly (~10 days/deployment) on a biannual basis by locally-recruited people operating from canoes. The personnel were not 'professional' staff from a vector control department as at the time of the intervention the vector control staff of the PNLTHA was comprised of only two entomologists, both of whom were based in Kinshasa. As a consequence of this capacity gap, vector control teams were entirely made up of people recruited from communities in the intervention area and using canoes hired from local boat owners. Our work in DRC shows that Tiny Targets provide a method of tsetse control that can be implemented by local people with a small (~1 day) amount of training and ongoing supervision. Moreover, Tiny Targets exerted their

greatest impact in areas close (<500m) to where they were deployed. This local effect is likely to strengthen the public perception that Tiny Targets have an impact. Their simplicity and obvious impact suggest that they might therefore be used in some form of community-based approach. However, rapid and regular deployment over large areas, including sections of sparsely inhabited rivers where tsetse are present, means that we caution strongly against the simple assumption that providing local communities with Tiny Targets and a minimum of training will lead to effective control of tsetse. In the present study, the project recruited and paid people from the villages, hired pirogues, provided instruction and supervision of target deployment, and monitored where targets were deployed and their impact on tsetse. These elements were essential to the success of the intervention. Further research is required to assess how local communities can best contribute to the effective use of Tiny Targets for effective and sustainable control of tsetse at scale.

## Conclusion

This study quantifies the impact of Tiny Targets on the tsetse fly population, demonstrating the dramatic and sustained decline in relative abundance immediately following the implementation of the intervention, accounting for factors such as habitat suitability and the distance to targets. The habitat suitability maps produced have the potential to be used to guide the expansion of activities in this area by both aiding in prioritising where to implement the intervention and determining the ideal placement of sentinel site monitoring traps to ensure the effect of the intervention can be monitored effectively.

## Supporting information

**S1 Table. Spreadsheet of the locations of the tsetse monitoring traps plus corresponding catch data, habitat suitability estimate and distance (m) to nearest Tiny Target.**
(XLSX)

**S2 Table. Environmental variables considered in the boosted regression tree and regularised regression models, plus their relative importance in final models (scaled out of 100).** The top five variables with regards to relative importance are shaded in green. Note that (^1) and (^2) in the LASSO column indicate whether the variable was considered as a linear or quadratic term in the model respectively.
(DOCX)

**S3 Table. GLMs and GLMMs fitted to the sentinel site data and their corresponding likelihood and AIC values.**
(DOCX)

**S1 Fig. Conditional modes of the cluster-level random effects obtained from the final fitted model.**
(DOCX)

## Author Contributions

**Conceptualization:** Steve J. Torr, Michelle C. Stanton.

**Data curation:** Inaki Tirados, Andrew Hope, Richard Selby, Fabrice Mpembele, Erick Mwamba Miaka, Steve J. Torr, Michelle C. Stanton.

**Formal analysis:** Michelle C. Stanton.

**Funding acquisition:** Erick Mwamba Miaka, Marleen Boelaert, Mike J. Lehane, Steve J. Torr.

**Methodology:** Michelle C. Stanton.

**Project administration:** Andrew Hope, Marleen Boelaert.

**Supervision:** Inaki Tirados, Andrew Hope, Richard Selby, Erick Mwamba Miaka, Steve J. Torr.

**Visualization:** Michelle C. Stanton.

**Writing – original draft:** Andrew Hope, Steve J. Torr, Michelle C. Stanton.

**Writing – review & editing:** Inaki Tirados, Andrew Hope, Richard Selby, Fabrice Mpembele, Erick Mwamba Miaka, Marleen Boelaert, Mike J. Lehane, Steve J. Torr, Michelle C. Stanton.

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
