## [Decision Letter · Decision Letter 0]

15 Jun 2020

Dear Dr Stanton,

Thank you very much for submitting your manuscript "Impact of Tiny Targets on Glossina fuscipes quanzensis, the primary vector of Human African Trypanosomiasis in the Democratic Republic of Congo." for consideration at PLOS Neglected Tropical Diseases. As with all papers reviewed by the journal, your manuscript was reviewed by members of the editorial board and by several independent reviewers. In light of the reviews (below this email), we would like to invite the resubmission of a significantly-revised version that takes into account the reviewers' comments. 

We cannot make any decision about publication until we have seen the revised manuscript and your response to the reviewers' comments. Your revised manuscript is also likely to be sent to reviewers for further evaluation.

Sincerely,

Enock Matovu

Associate Editor

Brian Weiss

Deputy Editor

Reviewer's Responses to Questions

**Key Review Criteria Required for Acceptance?**

**Methods**

-Are the objectives of the study clearly articulated with a clear testable hypothesis stated?

-Is the study design appropriate to address the stated objectives?

-Is the population clearly described and appropriate for the hypothesis being tested?

-Is the sample size sufficient to ensure adequate power to address the hypothesis being tested?

-Were correct statistical analysis used to support conclusions?

-Are there concerns about ethical or regulatory requirements being met?

Reviewer #1: The used methods are perfect

Reviewer #2: - Although the hypothesis is clearly stated, the objectives are not clearly defined. 

- The study design is appropriate to address the objectives 

- Statistical analyses are correctly performed

Reviewer #3: The objectives are clearly stated and the study design is appropriate to address the stated objectives. The population is clearly described and is appropriate for the hypothesis being tested, sample size sufficient,statistical analyses used to support conclusions and the study is ethical.

**Results**

-Does the analysis presented match the analysis plan?

-Are the results clearly and completely presented?

-Are the figures (Tables, Images) of sufficient quality for clarity?

Reviewer #1: The analysis presented matches the analysis plan

The results are clearly and completely presented

Tables and figures are of sufficient quality

Reviewer #2: - The results are not clearly and completely presented because entomological data are lacking

- The content of all tables are difficult to understand. see my comments below. The tables require foot notes and legends

- The legends of each figure must be revised

Reviewer #3: The results are clear, completely presented and tables and figures are clear.

**Conclusions**

-Are the conclusions supported by the data presented?

-Are the limitations of analysis clearly described?

-Do the authors discuss how these data can be helpful to advance our understanding of the topic under study?

-Is public health relevance addressed?

Reviewer #1: The conclusions are supported by the data presented in the manuscript. The manuscript is signifcant for the control of Gambian sleeping sickness

Reviewer #2: - The conclusions are not supported by the data presented because the result section does not provide enough entomological data.

- Results are of public health importance

Reviewer #3: The conclusions are supported by the data presented, with limitations clearly stated. The authors have discussed how their data can be useful to Tsetse and Trypanosomiasis control.

**Editorial and Data Presentation Modifications?**

Reviewer #1: 1. Reference is made to 1,234 unique locations in this manuscript as well as s1 T=[table 1]. It would be good for the authors to explain what was unique about these locations and how they were selected for inclusion in this study

2. It would very much help the readers if the authors produced raster [5 km resolution] maps using average FTD data for each trap for the periods; before the intervention; 0-2 months, 3-5 months and >5 months. These visual rendering would help the readers be able to palpate the effect of tinny targets by assessing if; the targets suppress tsetse populations across the entire terrain of the intervention area of if they suppress tsetse population and they shrink into small hot-spot tsetse subpopulations. 

3. The authors should consider the possibility of including a discussion point and or a simulation of how many intervention months are required for tinny targets to eliminate tsetse [at least for starters they can simulate isolated tsetse populations] in such a setting and if it is possible they could simulate non- isolated tsetse population with re-infestation prevented by barriers. This suggestion is discretionary!

Reviewer #2: (No Response)

Reviewer #3: Accept

**Summary and General Comments**

Reviewer #1: This is a very good paper in its field; only a few minor changes suggested above could further improve the paper for the readership of PNTD

Reviewer #2: This manuscript numbered PNTD-D-20-00527of Tirados et al entitled “Impact of Tiny Targets on Glossina fuscipes quanzensis, the primary vector of Human African Trypanosomiasis in the Democratic Republic of Congo” highlighted the use of Tiny Targets could also be used to accelerate the elimination of persistent foci of g-HAT in DRC. 

Major comments

This is an interesting study with the aims of developing approaches to implement large-scale trial of Tiny Targets in foci where vector control activities have been predicted to accelerate progress towards the elimination goal. These approaches will also assist in the designing and monitoring of the tsetse control operation through the production a map of tsetse habitat suitability. However, some major concerns related to this manuscript can be raised: 

- Although the hypothesis is clearly stated, the objectives are not clearly defined. 

- The study sites are not well described. As a vector-borne diseases and targeting the vectors, it will be important to provide some biotic and abiotic factors characterizing the biotopes where the traps were put during pre-intervention and the monitoring. 

- No real entomological data is presented in this manuscript. 

- It is mentioned that “The sex and species of collected tsetse were identified morphologically and then stored in 100% ethanol. Subsamples were subsequently analysed at the Liverpool School of Tropical Medicine (LSTM) to confirm species.” There is no data on sex and tsetse species. Where are results of analyzes performed at LSTM? How many flies were analyzed at LSTM? 

- How many flies were collected during the different phases? The authors mentioned that vector control was performed the first year of the study, and the operations were expanded subsequent years. The results section do not provide data related to these control operation: number of flies captured et each time, and the reduction rates

- On which criteria the 1,234 unique locations were selected for pre-intervention surveys of tsetse flies? How many flies were collected? These unique locations are not appearing on Figure 1. What did the authors refer to exploratory in the legend of figure 1? What do they mean by started (legend of figure 1)? Do they refer to areas where the tiny targets were deployed? 

- Why in some parts of figure 1, the exploratory of different years occurred in the same place? 

- What are the links between pre-intervention and the deployment of Tiny targets? Do results of pre-intervention operations guided the deployment of Tiny targets?

- Some legends of figure 2 are not appearing in this figure. 

- The legend related to tsetse count of figure 2 is not relevant. It can be seen as presence or absence of tsetse. It will be more convenient to present the relative density of tsetse flies

- From legends of figure 2, different estimated habitat susceptibility have been identified. What are the main characteristics of each habitat? 

- The authors mentioned in the discussion that “This low apparent density is unexpected given that the high incidence of g-HAT in Yasa Bonga [4]”. They tried to link tsetse density with the disease prevalence in humans. However, they do notremember that the transmission of trypanosomes does not depend entirely of tsetse density, but on the contact frequency between tsetse and vertebrate hosts. The tsetse density may also depend on the availability of vertebrate hosts. These two parameters have not been mentioned in this study.

- It is mentioned in the results section that “At baseline, there was no significant difference in the number of flies caught in traps set in areas with differing intervention status i.e. areas within the intervention area (‘in’), well outside of the intervention area (‘out’) and intermediate areas (‘edge’).” Where are these data?

- It is also mentioned that “Habitat suitability is significantly and positively associated with fly catches prior to the introduction of the intervention”. What type of habitat are the authors referring to? 

- Data of all tables are difficult to understand. The authors must give exactly what the data of each column are referring to. They must also include the foot notes for each table

With all problems mentioned above, this manuscript cannot be accepted for publication in Plos NTDs.

Reviewer #3: The manuscript has good research and has been well articulated. I have just a few recommendations to the authors:

1. It should be indicated that much as riverine tsetse is in west and central Africa, it is also present in some parts of East Africa, particularly the Lake Victoria and River Nile Basin. 

2. the findings about seasonal tsetse distribution and abundance need to be discussed in relation to previous studies.

PLOS authors have the option to publish the peer review history of their article (what does this mean?). If published, this will include your full peer review and any attached files.

Reviewer #1: Yes: Dr Muhanguzi Dennis [PhD], Makerere University College of Veterinary Medicine, Animal Resources and Biosecurity [COVAB]

Reviewer #2: No

Reviewer #3: No
---

## [Editor Report · Decision Letter 1]

26 Aug 2020

Dear Dr Stanton,

We are pleased to inform you that your manuscript 'Impact of Tiny Targets on Glossina fuscipes quanzensis, the primary vector of Human African Trypanosomiasis in the Democratic Republic of Congo.' has been provisionally accepted for publication in PLOS Neglected Tropical Diseases.

Best regards,

Enock Matovu

Associate Editor

Brian Weiss

Deputy Editor

---

## [Editor Report · Acceptance letter]

9 Oct 2020

Dear Dr Stanton,

We are delighted to inform you that your manuscript, "Impact of Tiny Targets on Glossina fuscipes quanzensis, the primary vector of Human African Trypanosomiasis in the Democratic Republic of Congo.," has been formally accepted for publication in PLOS Neglected Tropical Diseases.

Best regards,

Shaden Kamhawi

co-Editor-in-Chief

Paul Brindley

co-Editor-in-Chief
